# Postnatal depression and its social-cultural influences among adolescent mothers: A cross sectional study

Chimwemwe Tembo[1]*, Linda Portsmouth[2], Sharyn Burns[2]

**1** Saint John of God Hospitaller Services Malawi, Mzuzu, Malawi, **2** School of Population Health, Faculty of Health Sciences, Curtin University, Perth, Australia

* chimweptembo@yahoo.co.uk

## Abstract

In rural Malawi, adolescent mothers represent 31% of pregnancies. While some adolescent mothers experience motherhood as an exciting, positive, and affirming experience, for others, it may increase their risk of postnatal depression (PND). Social norms and culture contribute to adolescent mothers' experiences and may influence their mental health. However, there is limited research around the prevalence of PND and its cultural influences among adolescent mothers. A cross-sectional survey was administered from 7th September 2021 to 31st March 2022. Three hundred and ninety-five adolescent postnatal mothers aged ≤19 were conveniently recruited at Mitundu hospital in Lilongwe, Malawi. The Edinburgh Postnatal Depression Scale was used to assess depression. A cutoff point of ≥ 10 was employed to categorize probable PND. Binary logistic regression was used to determine the predictors of depression. The mean age of participants was 17 (SD 1.157). and 43.6% (n = 172) presented with PND (EPDS scores ≥10). When all factors were considered in a binary logistic model, adolescents who had ever experienced intimate partner violence (IPV) were 13.6 times more likely to report PND after controlling for age and other predictors compared to those without an experience of IPV (aOR 13.6, p = 0.01, 95% CI 2.10–88.9). Participants whose families did not decide for them (regarding their care) were 2.3 times more likely to present with PND than those whose families decided for them (aOR 2.3 p = 0.03, 95% CI 1.04–5.2). Adolescent mothers who had interacted with their health worker were less likely to report PND than those who had no interaction with the health worker (aOR 0.4 P0.02, CI 0.17–0.67). Social and cultural factors can impact mental health of adolescent mothers. It is recommended that targeted and integrated interventions are developed, implemented. and evaluated. There is a need to improve policy and practice to better support adolescent mothers postnatally.

## Introduction

Adolescent motherhood is a serious public health concern. Globally, 14 percent of adolescent girls give birth before turning 18 years old [1]. While in low-to-middle income (LMI) countries, it is estimated that about 12 million girls between the ages of 15 and 19 give birth each

**Data Availability Statement:** Data are hosted by Curtin University survey office through the Human Research Ethics Committee for researchers who meet the criteria for access to confidential data.

Interested researchers may reach out to the Human Research Ethics Committee (hre@curtin. edu.au), or the chairperson Human research ethics committee.

**Funding:** The research is part of a doctoral scholarship awarded by Curtin University to the corresponding author. Therefore, funding was provided to CT by Curtin University and the Malawi National Commission for Science and Technology to assist with data collection. The funders had no role in study design, data collection and analysis, decision to publish, or preparation of the manuscript.

**Competing interests:** The authors have declared that no competing interests exist.

year [2]. Currently, in rural Malawi, adolescent mothers contribute to about 31% of births [3, 4]. Malawi falls into the top ten nations when considering rates of adolescent pregnancies in Sub-Saharan Africa [5]. Most births to adolescent mothers occur among those from low social economic status, who have less education, and who live in rural areas [5].

The majority of adolescent pregnancies are unintended [6]. Parenthood presents challenges to women of any age; however, postnatal adolescent mothers face a unique range of challenges. They are more likely to experience a greater range of life stressors compared to adult mothers [7, 8]. Inadequate support from partners, friends, and family, can compound these challenges [8, 9]. It is hypothesized that these factors put postnatal adolescent mothers at a higher risk of common mental health disorders such as depression compared to older women [10, 11]. Research has found adolescent mothers who are parents to be twice as likely to experience common mental health disorders compared to mothers aged over 18 years and non-parenting adolescents [9, 10, 12]. The postnatal period is recognized as a time when women are at higher risk of perinatal mental health problems [13]. Post-natal depression is estimated to impact approximately 17% of mothers globally, however there are estimates of 20% among women in LMIC [10, 14–17].

In Malawi, a study conducted at Thyolo District Hospital among mothers attending the child health clinic [mean age 24.4 years] with infants aged 0–9 months [n = 501] found 30.4% of participants had experienced a minor or major depressive episode, with 13.9% currently experiencing a major depressive episode [18]. However, to the authors' knowledge, there are no previously published studies describing rates of depression for adolescent mothers in Malawi.

While, for some adolescent mothers, parenting is an exciting, positive, and affirming experience [19, 20] for others, there are a number of risk factors that increase the likelihood of negative psychological outcomes. Adolescent mothers may face an increased burden of responsibility related to childbirth, care of their baby, breastfeeding, and economic strain [10, 19]. Many adolescent mothers lack practical parenting skills, and are less prepared for pregnancy and motherhood than older mothers [21]. For some, parenthood may increase the risk of depression [19, 22]. Untreated postnatal depression is associated with a range of issues, including: increased incidence of poor maternal attachment [23]; suicide; infanticide; and increased risk of developmental difficulties, behavioral problems, conduct disorders, stunting, and lack of school enrolment for the child [5, 19, 24].

Psychosocial challenges experienced by adolescent mothers, impact negatively on help seeking and treatment for depression. For example, lack of mental health services, the stigma associated with mental health issues, and a lack of knowledge coupled with cultural beliefs, impede access to mental health care services [25]. This is particularly true in Malawi, where delays in seeking help are common among young mothers [26].

Culture is an important context for most experiences, shared beliefs, attitudes, and norms for emotional response and therefore affects how individuals experience mental health disorders, and the need to seek help [25]. It has been suggested that, in resource-constrained countries, women are protected from experiencing perinatal mental problems through the influence of social and traditional cultural practices during pregnancy and in the postpartum period [27, 28]. For example, in Pakistan [29] and Vietnam [28], social support from caring parents and adults within the community or family can be a protective factor for adolescents and could compensate for the absence of other protective factors and promote resilience [27, 30]. However, some residual cultural traditions, norms, and taboos enhance adolescents vulnerability to mental health issues [30]. Such cultural norms may act as barriers to interventions. In Malawi, cultural norms, practices, and taboos vary between tribes; hence, rather than generalizing, interventions specific to each ethnic group are required [31].

Cultural challenges experienced by adolescent mothers in Malawi are not well researched. Most research has focused on older mothers [32, 33], despite international evidence of adolescent mothers' vulnerability and the need for culturally appropriate interventions. This paper describes the prevalence and social and cultural influences of depression among adolescent mothers attending community clinics at Mitundu in Lilongwe, Malawi. These findings are part of a broader mixed- methods study [34]. The results inform the policymakers to consider screening adolescent mothers for mental health disorders during the postnatal period as part of a possible appropriate intervention to improve identification and management.

## Methods

### Ethics statement

The research is governed by the principles of beneficence, justice, and respect and is in accordance with the ethical principles of the Declaration of Helsinki [34]. Ethics approval was granted, recognizing adolescent mothers as emancipated minors; with participants under the age of 18 being asked additional questions to ensure they understood their potential involvement in the study. Participants who were illiterate and not able to sign their names used their thumbprints as a signature [this is standard practice in Malawi]. All participants provided informed consent. Where necessary, legal guardians witnessed the adolescent mother's consent [34], and ethical approval was received from the Curtin University Human Research Ethics Committee [HRE2021-0223] and the Malawian Ethics Board National Committee on Research Ethics in the Social Sciences and Humanities [P.05/21/575]. Any woman who reported suicidal ideation was counselled and offered a referral for clinical mental health assessment and treatment. Participants who were distressed were offered referrals for professional counselling and contacts for available tele- psychological counselling services.

### Study design and setting

A cross-sectional survey was conducted from 7th September 2021 to 31st March 2022, to adolescent postnatal mothers who had given birth within 11 months and came for Maternal and Child Health services at Mitundu Rural Hospital (MRH) clinics. The hospital is situated to the south-east of Lilongwe City. The catchment of MRH is characterized by high levels of poverty, high rates of early marriages, and malnutrition, and 90% of the population are subsistence farmers [ICF, 2017]. Mitundu has a population of 147,822 people, 29,171 households, and 402 villages. About 48% of the population is under 15 years of age. The catchment area of Mitundu is dominated by the Chewa tribe, which comprises 34% of the Malawian population [4]. The district adolescent fertility rate for women between the ages of 14–19 is 165 per 1,000 women, with an average of 135 adolescent deliveries per month. Twelve percent of women have never attained any education, and 62% have only completed primary school [4]. Mitundu has one mental health nurse who manages mental health services. Mental health services are supported by Saint John of God Hospitaller Services, Malawi. MRH provides free outpatient services, under-five services, family planning services, immunizations, antenatal, labor, and postnatal services.

### Study population and sampling

Convenience sampling was employed to recruit participants. Adolescent mothers were invited to participate in the study when they presented at the MRH for postnatal check-ups, family planning, and under-five clinics. Inclusion criteria included being ≤19, having the capacity to consent, and residing within the Mitundu catchment area at the time of the study. To calculate

the sample size for the study, the Slovin sample size determination formula $n = N/[1+N^*[e]2]$ was used, where n was the sample size, N was the population size, and e was the level of precision [also called the sampling error] [35]. In this study, N was the projected population of adolescent mothers who were expected to deliver at Mitundu Hospital in 2021. According to data from the Lilongwe District Hospital Office, 30,653 pregnant women (of all ages) were expected to deliver at the hospital in 2021. From this population, 43.2% were likely to be adolescent mothers (source), and the level of precision (e) is 5 percent at a 95 percent confidence level, resulting in a minimum study sample size of 395.

## Procedure for data collection

Nurses and community workers distributed flyers and information sheets and informed adolescent mothers of the study. Flyers and information sheets were also placed in strategic areas of the hospital including, postnatal wards, under-five clinics, the family planning department, and outreach clinics. Site health workers introduced the researcher and trained research assistants (RAs) to the mothers at under-five, postnatal, and outreach clinics on clinic days. Health Surveillance Assistants (HSAs) (community health field workers) supported recruitment via outreach clinics. An RA was available to read the information sheet and consent form in Chichewa to potential participants who had poor reading literacy. The survey was administered verbally by trained RAs in a room within the clinic or under a nearby shelter. Responses were recorded electronically through Kobo Collect (the android -based application used to collect data that feeds into the toolbox). The interview took approximately 45 minutes to administer. Participants were provided with K3500 to thank them for their time.

## Data collection tool and measures

The survey comprised four sections, including 1) demographics and social factors; 2) behavioral factors; 3], the Edinburgh Postnatal Depression Scale (EPDS); and 4) questions about common cultural factors. Demographics included: age; socioeconomic status; the number of deliveries; tribe; occupation; education; marital status; type of marriage (polygamy or monogamy), and planned or unplanned pregnancy. Social factors included perceived social support, which was assessed using the validated Multidimensional Scale of Perceived Social Support (MSPSS) [36]. Socioeconomic status was measured using the Wealth Index of the study population [37]. The Wealth Index is a measure of household socioeconomic status in LMI countries based on the household's ownership of consumer goods, dwelling characteristics, type of drinking water source, toilet facilities, and other characteristics.

Section two included questions related to social behaviours including intimate partner violence (IPV); alcohol consumption; and tobacco use. Alcohol consumption was assessed using AUDIT-C. which is an internationally recognized brief alcohol screening tool [38]. The tool assesses alcohol consumption by asking three questions focusing on the frequency and level of alcohol consumption. AUDIT-C initially asks participants if they have consumed a standard drink of alcohol in the past 12 months. Participants were then asked how often they drink alcohol; how many drinks they consume on a typical day when they are drinking, and how often they have six or more drinks on one occasion. A cut-off point of 8 was used to categorise low risk and hazardous/harmful drinkers into a binary variable [38]. Participants were asked if they currently use or smoke any form of tobacco [yes or no response]. The Hurt, Insulted, and Threatened, screamed at and Sexually Abused [HITS] Scale was used to assess experience of IPV [39]. HITS includes five items assessing five types of abuse experienced during the past 12 months: being hurt, physically insulted, threatened, screamed at, and forced into sexual activity by their intimate partner (five point Likert scale; sores 0 to 5; range 0–25) [40]. A cut-off point

of ≥10 was employed to signal the presence of abuse [40]. The HITS has been used previously in the Malawi Demographic Health Survey, with the questions adapted to reflect the Malawian context [ICF, 2017].

Section three included questions relating to depression. The EPDS, a ten-item scale for diagnosing psychological symptoms of depression experienced in the past seven days, was employed. Questions are based on the DSM depression diagnostic criteria (Ten items 0–3, score range 0–30) [33, 41, 42]. As consistent with other studies in Malawi mothers were provided with visual prompt cards to facilitate responses [43], Responses were represented by 'emoji' or emotional faces that represented 'sad face', as "not at all" and 'happy face' meaning "as much as you always could". The frequency questions included groups of faces with varying proportions of 'sad' and 'happy' faces. The EPDS tool is widely used either as a population-based self-report or as a health-worker administered measure in Malawi [33, 36]. The EPDS has been validated in the predominantly rural population of perinatal mothers in Mangochi—Malawi with the internal validity found to be high (sensitivity index: r 68.8% - 100%; specificity index: 79.5: reliability: Cronbach's alpha 0.9] [32]. While the EPDS was designed as a self-completion tool, it has also been administered by interviewers [41, 42]. A cut-off of ≥10 was used to indicate probable postpartum depression, as consistent with other studies in sub-Saharan Africa [42, 44] and globally for adolescent mothers [45].

New questions were developed to explore the influences of mental health from a cultural perspective. These questions were informed by the literature to meet a gap and underwent construct, content and face validity testing, and a yes or no response option was provided (see Table 2).

## Translation of survey questionnaire

The survey underwent construct and content validity with three Malawian mental health experts, three public health specialists, two senior researchers, and two health workers. In addition, these professionals reviewed for face validity, to ensure the translation and wording of all questions were appropriate for local participants [46]. Following the World Health Organisation standardized and validated translation process [47], the survey was translated from English to Chichewa and then back by an independent professional translator. The face validity of the survey was achieved via expert review in addition to a focus group discussion (FGD) with 15 adolescent mothers. Language and comprehension of questions were explored during the FGD [46]. All questions were administered in Chichewa. In the process of translation, inadequate expressions or concepts and discrepancies were discussed and resolved during the face validation process.

## Data analysis

After data collection through Kobo Toolbox, the dataset was imported into SPSS version 27 for descriptive and inferential analysis. The dependent variable or outcome was probable PND with a binary outcome (probable PND and Low risk PND). Descriptive statistics were computed to describe the frequencies of independent variables of demographic, social, behavioural and cultural characteristics such as age, gender, educational level, marital status, use of alcohol, whether the child was planned or unplanned, place of delivery, religion, experience of loss, and social economic status. Social-economic status was computed as Wealth Index, which is a measure of the economic status of a household. The Principal Component Analysis (PCA) methodology was used to compute the wealth index.

To establish the prevalence of PND, the study performed descriptive analysis using frequency tabulations. In particular, PND was tabulated into a binary variable with a cut-off

point of, $\geq$10 which was considered probable PND. Furthermore, to determine the association between demographic, social and economic factors, a Pearson Chi Square test of association for categorical response variables was conducted. An independent t test was performed to obtain an association between probable PND and MPSS scores. Lastly, we conducted a binary logistic regression analysis to examine the relationship between a binary outcome variable (PND or no depression variable) and the predictor variables to measure the direction and magnitude of factors associated with postnatal depression. The covariates included were all significant factors identified via chi-square test of association and t-test, such as age, marital status, polygamy, age at marriage, occupation, place of delivery, MSPSS, wealth index, family freedom, being isolated, interaction with health worker, forced marriage, husband support, being prayerful, experience of loss, and IPV. However, HIV serostatus, past psychiatric history and alcohol use are considered as confounders in the literature, but these, were not included in the final logistics model because, they were not significant associated with probable depression in the bivariate analysis. Odds Ratios and Coefficients were used to determine the magnitude and direction of the association respectively. Statistical significance was determined at p<0.05. Inferential analyses included confidence level of 95% based on 2-tailed tests. Our analysis showed that after adjusting for the potential confounding variables, the predictor variables remained significantly associated with the outcome variable.

## Results

### Demographic and social characteristics of the participants

Four hundred and one eligible adolescent mothers agreed to consider participation in this study. After having the information sheet read to them, six opted not to participate. Two declined because they believed the researchers were from satanic organizations and may convert them as current Christians to Satanism. Two adolescent mothers were afraid of the conspiracy theories that surrounded the COVID-19 vaccine and believed the researchers would force them to receive this vaccine. Two participants declined because their husbands told them not to take part. The final sample was 395, representing a 99.5% response rate.

Participants' demographic and social characteristics are described in Table 1. The participants mean age was 17 (SD 1.157); 34% (n = 135) were aged 13–17 years, and 65% (n = 260) were 18–19 years. The majority of participants were married 78% (n = 308); twelve percent were in a polygamous marriage, and 73% were married before the age of 18. Most participants had given birth to their first child 90.6% (n = 358). Over 60% of participants had not planned their current pregnancy (63.3%, n = 250). Eighty-seven percent of participants had attended primary school while 10% (n = 40) had attended secondary school. Ninety-five percent of participants were Christians, and the majority belonged to the Chewa tribe 98%, (n = 387). The average MPSS score was 31. 17(SD± 10). Forty-two percent were within the first and second poorest quintiles and 20% were in the wealthiest quantile. IPV had been experienced by 5.3% (n = 21) of participants. The prevalence of probable postnatal depression (EPDS scores $\geq$10) tabulated among adolescent mothers was 43.5%, (n = 172).

Table 2 summarizes the possible cultural influences of mental health as experienced by the participants. The majority of participants (83.2% n = 322) reported a lack of interaction with any healthcare provider during the postnatal period. Fifty four percent, of the respondents (n = 212), reported they initially seek help from religious leaders to treat their mental health problems; and 255 (65.4%) sought help from a traditional healer. Almost one third of participants (31.8%, n = 124) had become more prayerful during their pregnancy.

When asked about their family influences, fifty-three percent of participants (n = 206) indicated they did not have the freedom to discuss their mental health with their family. Thirty

**Table 1. Demographics, social, behavioral, and emotional characteristics of participants (n = 395).**

| Variable | Frequency | Mean±SD or %/Percentage (%) |
|---|---|---|
| **Age (years)** | | |
| 13–17 | 135 | 34.1% |
| 18 above | 260 | 65.9% |
| **Marital status** | | |
| Single | 65 | 16.5% |
| Widow | 1 | 0.3% |
| Married | 308 | 78.0% |
| Divorced | 21 | 5.3% |
| **Polygamous marriage** | | |
| Yes | 38 | 12.3% |
| No | 270 | 87.7% |
| **Level of Education** | | |
| Never | 5 | 1.3% |
| Primary | 350 | 88.6% |
| Secondary | 40 | 10.1% |
| **Age at Marriage (years)** | | |
| 13–17 | **226** | 68.4.% |
| 18 above | **104** | 31.6% |
| **Occupation** | | |
| Farmer | 363 | 92.0% |
| Student | 17 | 4.3% |
| Business worker | 15 | 3.7% |
| **Number of live deliveries** | | |
| Once | 358 | 90.6% |
| Two | 35 | 9.0% |
| Three | 2 | 0.4% |
| **Planned pregnancy for the current child** | | |
| Yes | 145 | 36.7% |
| No | 250 | 63.3% |
| **Place of delivery** | | |
| Hospital | 366 | 92.7% |
| Traditional birth Attendants/Home | 29 | 7.3% |
| **Religion** | | |
| Muslim | 4 | 1.0% |
| Christian | 378 | 95.7% |
| Traditional Religion | 13 | 3.3% |
| **Tribe** | | |
| Chewa | 387 | 98.0% |
| Yao | 3 | 0.8% |
| Lomwe | 2 | 0.5% |
| Ngoni | 3 | 0.8% |
| **Who mother lives with** | | |
| Parents | 89 | 22.5% |
| Husband | 306 | 77.5% |
| **HIV status** | | |
| Positive | 5 | 1.3% |
| Negative | 347 | 87.8% |

*(Continued)*

**Table 1.** (Continued)

| Variable | Frequency | Mean±SD or %/Percentage (%) |
|---|---|---|
| Unknown | 43 | 10.9% |
| Multidimensional Scale of Perceived Social Support (MSPSS) | 395 | 31.17±10.4 |
| **Wealth index quintile** | | |
| 1st (poorest) | 68 | 17.4% |
| 2nd | 98 | 25.1% |
| 3rd | 68 | 17.4% |
| 4th | 78 | 20% |
| 5th (wealthiest) | 78 | 20% |
| **Experience loss** | | |
| Yes | 131 | 33.1% |
| No | 264 | 66.9% |
| **Ever had a full serve of alcohol in the last 12 months** | | |
| Yes | 8 | 2.0% |
| No | 387 | 98% |
| **Alcohol consumption (AUDIT C) (Scores cut off point <8)** | | |
| Low level | 2 | 25% |
| Hazardous / Harmful level | 6 | 75% |
| **Ever used or smoke any form of tobacco in the past 12 months** | | |
| No | 395 | 100% |
| **Ever experienced Intimate Partner Violence (IPV) using HITS scores cut of ≤ 10** | | |
| Yes | 21 | 5.3% |
| No | 374 | 94.7% |
| **Overall prevalence of self-reported depression (cut point of ≥10)** | | |
| Probable PND | 172 | 43.5 |
| low risk of PND depression | 223 | 56.5 |

seven percent (n = 147) experienced a negative reaction from their parents due to their pregnancy. More than half (77%, n = 302) of the participants reported that their family members decide on their behalf regarding their care, and the majority (82.6%, n = 322) received information regarding childbirth and parenting from community elders. Almost three quarters (74%, n = 289) of participants reported that they are financially supported by their husbands.

During pregnancy and the postpartum period, 86.4% (n = 337) of participants reported following some strict cultural practices related to childbirth. For example, having traditional ceremonies regarding preparation for childbirth, and parenting 82.6% (n = 322). Additionally, 10.8% (n = 42) reported being forced to have sex with a man who was not their husband due to traditional cultural beliefs.

## Demographic, social and cultural factors associated with probable PND among adolescent mothers during the postpartum period

Demographic factors significantly associated with probable PND include: age, whereby older participants (aged 18–19 years) were more likely to report probable PND symptoms compared to younger participants (13–17 years) (63.8%, $X^2$ = 14.591, <0.001); being single (63.1%, $X^2$ = 17169, p = 0.001) compared to being married; those in polygamous relationships (60.5%, $X^2$ = 9.051 p = 0.004) compared to those in a monogamous marriage; adolescents who delivered at traditional birth attendants were more likely to report PND symptoms (76%, $X^2$ = 8.767 P = 0.003) compared to those who delivered in a hospital (Table 3). In terms of support,

**Table 2. Cultural influence factors.**

| Factors | Yes f[%] | No f[%] |
|---|---|---|
| Seeking help from traditional, spiritual healers first before hospital | 255 (65.4%) | 135 (34.6%) |
| Seek help from religious leaders first to treat mental health problems | 212 (54.4%) | 178 (45.6) |
| You have become more prayerful to seek God's help since delivery. | 124 (31.8%) | 266 (68.2%) |
| Freedom to discuss mental health and baby care within family. | 184 (47.2%) | 206 (52.8%) |
| Positive reaction by adolescent mothers' parents towards the gender of the baby | 301 (77.2%) | 89 (22.8%) |
| Negative reaction to pregnancy by immediate family and other relatives | 147 (37.7) | 243 (62.3%) |
| Being isolated during family activities because you have a baby | 46 (11.8%) | 344 (88.2%) |
| Family making decisions on your behalf regarding your care | 302 (77.4%) | 88 (22.6%) |
| Having a child before marriage is unacceptable | 301 (77.2%) | 89 (22.8%) |
| Forced to get married to someone | 112 (28.7%) | 278 (71.3%) |
| Being supported by husband | 289 (74.1%) | 101 (25.9%) |
| Arranged traditional ceremonies regarding childbirth, and parenting that are organized by family or community elders | 322 (82.6%) | 68 (17.4%) |
| Strict cultural rules beliefs and practices regarding pregnancy and childbirth for example specific rituals to follow | 337 (86.4) | 53 (13.6%) |
| Experiencing restrictions to cooking and socializing because of pregnancy & delivery | 218 (55.9%) | 172 (44–1%) |
| Being forced to have sex with a man because of cultural practices | 42 (10.8%) | 348 (89.2%) |
| Interaction with health care providers during pregnancy and post delivery | 66 (16.9%) | 324 (83.1%) |

participants who had more support were more likely to report PND scores <10 compared to those with limited support (t = 8.910, p = 0.001). Participants from the poorest and wealthiest indexes were more likely to report EPDS scores of $\geq$ 10 compared to those from the second, third and fourth wealth index (P = 0.001). Participants who reported a lack of family freedom to discuss issues were more likely to report PND symptoms scores of $\geq$10 (55.3%, 24. 650[a] p = 0.001). Adolescents who experienced a negative reaction about their pregnancy from their parents (52.7%, $X^2$ = 8.186a P = 0.004), were isolated within their family (86%, $X^2$ = 10.901[a], p = 0.001), whose family does not make decisions on their behalf (57.1%, $X^2$ = 8.894[a] p = 0.003), reported a lack of interaction with health workers (63.6%, $X^2$ = 13.012a p = 0.001), and were in forced marriages (54.4% $X^2$ = 7.662[a], p = 0.006), were significantly more likely to report PND symptoms scores of $\geq$10 (Table 3).

## Factors associated with postnatal depression in a binary logistic regression

Adolescents who ever experienced IPV were 13.6 times more likely to report postnatal depression than those without experience of violence (aOR 13.6, p = 0.01, 95% CI 2.10–88.9). Being prayerful reduced the odds of presenting with postnatal depression (aOR 0.4 p = 0.02, CI: 0.2–0.87). Those whose families were not making decisions for them regarding their care were 2.3

**Table 3. Factors associated with probable postnatal depression (EPDS ≥ 10).**

| Variable | Low risk of PND <10 EPDS score | Probable PND ≥10 EPDS score | $X^2$ statistic/t-test | P value |
|---|---|---|---|---|
| **Demographic Factors** | | | | |
| **Age** | | | | |
| 13–17 | 75(56.4%) | 58(43.6%) | 14.591 | <0.001** |
| 18–19 | 93(36.2%) | 164 (63.8%) | | |
| **Marital status** | | | 17.169 | 0.001** |
| Single | 24 (36.9%) | 41 (63.1%) | | |
| Widowed | 1 (100%) | 0 (0%) | | |
| Married | 190 (61.7%) | 118 (38.3%) | | |
| Divorced | 8 (38.1%) | 13 (61.9%) | | |
| **Polygamous marriage** | | | 9.051 | 0.004** |
| Yes | 15 (39.5%) | 23 (60.5%) | | |
| No | 175 (64.8%) | 95 (35.2%) | | |
| **Age at marriage** | | | 8.094 | 0.004** |
| 13–17 | 96 (42.5%) | 130 (57.5%) | | |
| 18–19 | 19 (24.4%) | 59(75.6%) | | |
| **Occupation** | | | | |
| Full-time job | 24 (75.0%) | 8 (25.0%) | 4.871[a] | 0.027** |
| Home Duties | 115(71.4%) | 46 (28.6%) | 24784[a] | 0.001** |
| Subsistence arming | 194 (58.3%) | 138 (41.7%) | 2.804[a] | 0.09 |
| **Level of education** | | | 1.410[a] | 0.494 |
| None | 4 (80.0%) | 1 (20.0%) | | |
| Primary | 195 (55.7%) | 155 (44.3%) | | |
| Secondary | 24 (60.0%) | 16 (40.0%) | | |
| **Level of education of the spouse** | | | 4.337a | 0.114 |
| None | 18 (81.8%) | 4 (18.2%) | | |
| Primary | 136 (59.4%) | 93 (40.6%) | | |
| Secondary | 36 (63.2%) | 21(36.8%) | | |
| **Gender of the baby** | | | 0.127 | 0.72 |
| Male | 101 (55.5%) | 81 (45.5%) | | |
| Female | 122 (57.3%) | 91 (42,7%) | | |
| **Planned or unplanned pregnancy** | | | .058[a] | 0.81 |
| Yes | 83 (57.2%) | 62 (42.8%) | | |
| No | 140 (56.0%) | 110 (44.0%) | | |
| **Place of delivery** | | | 8.767 | 0.03** |
| Hospital | 213 (58.2%) | 153 (41.8%) | | |
| Traditional birth Attendants | 3 (23.0%) | 10 (76.0%) | | |
| Home birth | 6 (40.0%) | 9 (60.0%) | | |
| **Intimate partner Violence [HITS ≥ 10]** | | | 19.872 | 0.001** |
| No | 221 (59.1%) | 153 (40.9%) | | |
| Yes | 2 (9.5%) | 19 (90.5%) | | |
| **Economic factors associated with depression** | | | | |
| **Wealth index** | | | | |
| 1st quintile [poorest] | 20 (27%) | 54 (73%) | 43790a | <0.001** |
| 2nd quintile | 69 (65.1%) | 37 (34.9%) | | |
| 3rd Quintile | 16 (29.1%) | 39 (70.9%) | | |
| 4th Quintile | 42 (53.2%) | 3 (46.8%) | | |

*(Continued)*

**Table 3.** (Continued)

| Variable | Low risk of PND <10 EPDS score | Probable PND ≥10 EPDS score | $X^2$ statistic/t-test | P value |
|---|---|---|---|---|
| 5th Quintile [wealthiest] | 21 (27.6%) | 55 (72.4%) | | |
| Multi-dimensional scale of perceived social support [MSPSS] | | | 8.910 | <0.001** |
| Experience of loss | | | 4.686 | 0.03** |
| yes | 84(64.1%) | 47(35.9%) | | |
| no | 139(52.7%) | 125 (47,3%) | | |
| **Cultural factors** | | | | |
| **Seek help from religious** | | | 0.157[a] | 0.76 |
| Yes | 120 (55.6%) | 96 9 (44.4%) | | |
| No | 103 (57.5%) | 76 (42.5%) | | |
| **Family freedom to discuss mental health** | | | 24.650[a] | 0.001** |
| Yes | 130 (69.5%) | 57 (30.5%) | | |
| No | 93 (44.7%) | 115 (55.3%) | | |
| **Negative reaction to pregnancy by family and relatives** | | | 8.186[a] | 0.004 |
| Yes | 71 (47.3%) | 79 (52.7%) | | |
| No | 152 (62.0%) | 93 (38.0%) | | |
| **Being isolated in family activities because baby** | | | 10.901[a] | <0.001** |
| Yes | 16 (34%) | 31 (86%) | | |
| No | 207 (59.5%) | 141 (40.5%) | | |
| **Family making decisions on your behalf regarding your care** | | | 8.894[a] | 0.003** |
| Yes | 184 (60.5%) | 120 (39.5%) | | |
| No | 39 (42.9%) | 52 (57.1%) | | |
| **Lack of interaction with the health care providers** | | | 13.012[a] | <0.001** |
| Yes | 24 (36.4%) | 42 (63.6%) | | |
| No | 199 (42.9%) | 130 (39.5%) | | |
| **Forced to get married to someone** | | | 7.662[a] | 0.006** |
| Yes | 52 (45.6%) | 62 (54.4%) | | |
| No | 171 (60.9%) | 110 (39.1%) | | |
| **Being supported by husband \*** | | | 16.625 | 0.001** |
| Yes | 183 (62.5%) | 110 (37.5%) | | |
| No | 40 (39.2%) | 62 (60.8%) | | |
| **You have become more prayerful to seek God for help** | | | 26.445 | 0.001** |
| Yes | 176(65.2%) | 94(34.8%) | | |
| No | 47 (37.6%) | 78 (62.4%) | | |
| **Being forced to have sex with another man because of cultural practices** | | | 3.549[a] | 0.06 |
| Yes | 19 (43.2%) | 25 (56.8%) | | |
| No | 204 (58.1%) | 147 (41.2%) | | |

** Significant at p = 0.05, a = chi test result or t test result

times more likely to report postnatal depression than those whose families made the decisions for them (aOR 2.3 p = 0.03, 95% CI: 1.04–5.2). Adolescent mothers who had interacted with a health worker were less likely to be depressed postnatally than those who did not interact with the health worker (aOR 0.4 P0.02, CI: 0.17–0.67). There was an inverse relationship between support and postnatal depression. A one unit increase in multi-dimensional scale of perceived social support score resulted in an 11% decrease in postnatal depression scores.

**Table 4. Bivariate logistic regression analysis of factors associated with postnatal depression in adolescent mothers.**

| Variable | Odds Ratio | Coefficient [*B*] | P value | Confidence Interval |
|---|---|---|---|---|
| **Intimate partner violence (IPV)** | | | | |
| No | Reference | | | |
| Yes (Positive) | 13.6 | 2.116 | 0.01 | 2.10–88.9 |
| **Being prayerful** | | | | |
| Yes | Reference | | | |
| No | 0.4 | 0.923 | 0.02 | 0.2–0.87 |
| **Family making decisions on your care** | | | | |
| Yes | reference | | | |
| No | 2.3 | -.230 | 0.03 | 1.04–5.2 |
| **Interaction with health** | | | | |
| No interaction | reference | | | |
| Had interaction | 0.4 | -0.668 | 0.02 | 0.17–0.67 |
| **Multidimensional social support** | 0.9 | -0.912 | <0.001 | 0.8–0.9 |

1.0 Reference group.

Adjusted for all variables in the table as well as marital status, polygamy, age at marriage, occupation, place of delivery, forced marriages, the wealth index family freedom, negative reactions, feeling isolated, family decisions, not being supported, and forced sex.

In terms of the direction of association, there is a positive correlation between IPV and being prayerful with postnatal depression. While having more support, parents deciding on your behalf, and health worker interaction were inversely correlated with PND. [See Table 4].

## Discussion

This is the first study to determine the cultural factors associated with the prevalence of depression among postnatal adolescent mothers in Malawi. This is noteworthy as, while postnatal depression has been studied in adults, there is no research focusing on adolescent mothers in Malawi. The prevalence of probable PND among adolescent mothers in this sample was 43.5% using the EPDS cut -off point score of ≥10. This is relatively similar to the higher end of the reported global adolescent estimated prevalence of 25–42% [7, 9, 45, 48] and higher than the findings of two studies among postnatal adult mothers in Malawi within similar rural settings. These studies found a prevalence of 30.4% [18] and 19% [33]. A study in Rwanda found a 48% prevalence of postnatal depression among adolescent mothers in a rural setting using an EPDS cut off point of 13 [5], while a US study among African American adolescents found a prevalence of 45% using the Beck Depression Inventory II [49]. The findings of this study are similar to other studies that have found adolescent mothers to have an increased risk and higher prevalence of postnatal depression compared to adult mothers [7, 11, 19, 50, 51]. This suggests that adolescent mothers experience a range of different issues compared to adult mothers, which may include parental pressure, poor problem-solving skills, the inability to think abstractly [51, 52], psychological stress [8] which may be influenced by their inability to effectively cope with parenting, family dysfunction, and a lack of practical parenting skills [51, 53]. The findings of this study justify the consideration of clinical actions that can be employed at the primary health care level specifically for postnatal adolescent mothers.

Although prevalence estimates vary across countries, some psychosocial factors associated with depression are prevalent across high- middle- and low-income countries. For example, experiencing IPV, being single, a lack of social support, feeling isolated, and having a lower economic status, were found to be significantly associated with high depression scores in this

study. These findings support those of other studies conducted in the US, South Africa, and Nigeria among adult postnatal women [9, 12, 27, 48, 54, 55]. However, other factors such as level of education, [p = 0.49] unintended or planned child [p = 0. 81], and child gender [p = 0.72] were not significantly associated with depression in this study, but they were associated with depression in other studies [9, 12, 27, 49, 54, 56]. The differences in results may be attributed to homogeneity within the responses, which resulted in a lack of variability within the sample responses to 'education', and 'unintended or planned child'. This is because the majority of adolescents clustered on a response that was similar. For example, responses for the level of education. As for unplanned pregnancies the reason would be, because of social and cultural normalization of adolescent childbearing whereby, having a child in Malawi gives a woman an affirmation and a sense of recognition within society [57]. Therefore, when the child is born, having an unplanned child does not matter because the majority of them get married and they are supported by their husbands. In addition, the gender of the child does not matter in this society. Furthermore, this study was conducted in a rural setting, where the cultural context is more communal than individual. Hence, mothers receive significant community-level support, highlighting the importance of social support for new mothers. Therefore, future studies may want to compare rural and urban adolescents. However, adolescent mothers who are not married, are more likely to experience shame and stigma and may not be afforded the same support [24, 58]. Notably, in this study, adolescent mothers who were not married were more likely to report depressive symptoms.

The study found a negative correlation between MPSS and depression, where by a decrease in MPSS support resulted in an increase in depression, this is similar to many studies conducted among adult perinatal women [36, 44] An individual's ability to handle pressures is thought to be significantly improved by social support [44, 59]. Hence, adolescent mothers may have benefited from extra support.

Participants in the poorest and wealthiest quintiles were more likely to report postnatal depressive symptoms (scores of $\geq$ 10) compared to those within the 2nd 3rd and 4th. quintile. These results are surprising. Studies in low income countries such as Rwanda, high income countries such as the US and middle income countries such as Mexico found high socio economic status to be associated with a lower likelihood of depression during the postnatal period among adolescent mothers [5, 60–62]. The first author speculates that the finding may be attributed to transgenerational marriage, where there is sexual division of power and power imbalances within the relationships at play despite the adolescent mother living with her comparatively wealthy older husband [63, 64]. Adolescent mothers also likely lack decision making power with regard to financial and economic activities due to gender specific cultural norms where men become controllers of all finances [63] and women are restricted in undertaking economic activities after child birth as they are expected to be at home [65]. The lack of economic activity increases the odds of depression in the general population [66]. Ccontrary to the above commentary, another study in the US found household poverty was not a significant predictor of postnatal depression [67]. Therefore, there is a need for further investigation of this research outcome.

The child's gender did not significantly influence depression in this study, unlike the findings of studies in Turkey [68] and Uganda [69] where male children are preferred and thus associated with a reduced risk of depression. Malawi is a matrilineal society, where female and male children have equal value. Additionally, it is assumed a female child will one day 'own' the village because males marry and leave the village.

Level of education was not significantly associated with depression in this study. This is because in this sample, approximately 99% of participants had not attended secondary education. This has caused the responses to be clustered on primary school and there is a lack of

variability leading to education being not significant. Data from Malawi revealed that 72% of young people do not complete secondary school education [4].

This study found adolescent mothers whose family/parent family did not decide for them regarding their care were 2.3 times more likely to present with depression than those whose parents decided for them. A Rwandan study found dysfunctional parental interaction to be associated with postnatal depression among adolescent mothers [5]. This finding highlight that limited resilience, conflict resolution, and coping skills, coupled with the pressures of undertaking adult roles among adolescent mothers, are likely to impact depression. Furthermore, the results may indicate the effects of diminished communication with their parents, particularly when the pregnancy is not planned. Unplanned adolescent pregnancies in Malawi are associated with shame and defilement and may lead to diminished parental support [24]. In addition, in Malawi, if an adolescent falls pregnant, she is immediately forced into marriage or she is taken to the man who is responsible or to her in-laws' home to live, where in most cases, she is expected to make her own decisions regarding her care. Furthermore, there is a cultural norm or tradition of no communication about reproductive issues or sensitive topics between adolescents and their parents or mothers-in-law which may enhance the vulnerability of the adolescent mothers. This may result in adolescent mothers searching for other systems available within their culture to provide support. It is clear that other existing systems within the community, such as key members of the church, can be empowered with mental health information and basic counselling skills to enhance their support of young mothers.

This study found a positive correlation between being prayerful and postnatal depression. However, this is contrary to a study conducted in Pakistan among female adolescents in universities, which found a significant inverse association between religiosity variables and depressive symptoms among young women in the general population [70]. The author speculates that adolescent mothers with a strong commitment to religious life may have a stronger social network due to relationships that are fostered in religious settings and have a greater sense of belonging to a group [71]. Therefore, in situations when they face life challenges, they may become more prayerful to seek God's intervention'. In addition, in the absence of other services within the community, adolescents may use religious or faith pathways and become more prayerful to cope or deal with issues. Therefore, prayer, or being engaged in a religious ritual, is used to cope with stressors. Prayer may also be used to seek God's intervention to change their partners' behaviors if they suffer abuse from partners or significant others [71, 72]. Therefore, working with faith-based organizations and religious leaders in the community to create and evaluate mental illness prevention programs is essential.

Adolescent mothers who had interacted with the health worker were less likely to be depressed than those who had no interaction. This is the first study of adolescent mothers in Malawi that has demonstrated the correlation between depressive symptoms and contact with a health worker. Considering that the majority of adolescent mothers are primiparas, and the clinical implications of poor mental health, this finding justifies the need for clinical actions by health professionals to support adolescent mothers at the primary care level not only during pregnancy but in the postnatal period as well. Furthermore, Malawian maternal and child clinics would be ideal settings for screening and engaging with postnatal mothers.

None of the participants in this study reported tobacco use, and only 25 [n = 8%] reported consuming of alcohol during the postpartum period. However, of those who drank alcohol, most [n = 6; 75%] reported hazardous/harmful alcohol consumption levels. This is a public health concern considering the short term and long term harms associated with harmful and hazardous alcohol consumption [73, 74]. The low proportion of adolescent mothers reporting alcohol consumption in this study may be associated with changing social norms and the increased responsibility for new mothers protecting them against alcohol use [74]. Rates of

consumption are lower than findings from the US among teen mothers [9, 75] and South Africa among perinatal adolescents [76–79]. However, residual cultural norms are more prominent in Malawi, where alcohol and tobacco use are not socially acceptable among women and those who do consume these substances are considered promiscuous [80]. While these data may be representative for adolescent mothers in Malawi, given these cultural norms, social desirability bias in their responses is a possibility [81].

Similar to other studies, this study found IPV to be a predictor of depression [27, 42, 82–86]. However, compared to studies in other countries, the prevalence of IPV was lower [5.3%]. Studies have reported that 25% of new mothers globally experience IPV after childbirth [39, 87]. In Malawi, to the best of the first author's knowledge, there is a paucity of data describing IPV experiences among adolescent mothers during the postnatal period. However, the Malawi Demographic Health Survey [2016] found that 5% of women reported experiencing physical violence during pregnancy. The first author speculates that adolescent mothers may have feared their partners being imprisoned or being subjected to harsh penalties if they admitted experiencing violence. In addition, in this catchment area, there are initiatives by other Non-Governmental Organization to stop gender-based violence and early marriages, which may have resulted in underreporting. Another possible explanation proposed by the first author is that adolescent couples are mostly in smooth relationships with no power imbalance and threats are likely to be uncommon. In addition, increased social support following the birth of the child may be associated with a lower likelihood of IPV [8, 84]. IPV challenges the development of the child because it may foster the psychological unavailability of the mothers and may interfere with caregiving [8, 84]. Prevention initiatives such as promoting companionship in clinics provide an opportunity to provide information and enhance skills for couples around supportive relationships [88]. Notably, a study in South Africa found that father involvement can improve postnatal maternal depression and that men would benefit from specific guidance on how they can support mothers during and after delivery [89].

## Limitation

As this is a cross-sectional study, causal effects cannot be implied. In addition, we did not account for other potential confounders such as hormonal changes, and physical health problems that might have an effect on the mother [49]. However, the predictors highlighted in this study can inform advocacy initiatives for public mental health planning, and the development of health promotion interventions and strategies in Malawi. This study may also be subject to selection bias, as it was a clinic-based study using a convenience sampling method. However, to reduce the bias, postnatal adolescent mothers who presented at the clinic were randomly invited to participate in the study. Given the sensitive nature of some of the questions, social desirability bias may have impacted some findings. It is recognised that the EPDS is a screening tool and not a diagnostic but, it is widely used internationally to determine population prevalence [45, 90].

## Conclusion

The prevalence of PND among adolescent mothers is of public health concern. Some social and cultural factors are found to either have a negative influence on mental health or to protect mothers' mental health. Multidimensional Social support and professional health support were found to promote good mental health. However, forced marriages, delivering with a traditional birth attendant, and experience of IPV were associated with increased PND among adolescent mothers. It is strongly recommended that targeted and integrated interventions be developed, implemented, and evaluated utilising new models of care that employ a

multidisciplinary and collaborative approach to maternal care, particularly for adolescent mothers in Malawi.

## Acknowledgments

The authors acknowledge the support the Lilongwe District Health Office rendered during data collection.

## Author Contributions

**Conceptualization:** Chimwemwe Tembo, Linda Portsmouth, Sharyn Burns.

**Data curation:** Chimwemwe Tembo, Linda Portsmouth, Sharyn Burns.

**Formal analysis:** Chimwemwe Tembo, Linda Portsmouth, Sharyn Burns.

**Funding acquisition:** Chimwemwe Tembo.

**Investigation:** Chimwemwe Tembo, Linda Portsmouth, Sharyn Burns.

**Methodology:** Chimwemwe Tembo, Sharyn Burns.

**Project administration:** Chimwemwe Tembo, Sharyn Burns.

**Resources:** Chimwemwe Tembo, Sharyn Burns.

**Software:** Chimwemwe Tembo, Sharyn Burns.

**Supervision:** Linda Portsmouth, Sharyn Burns.

**Validation:** Linda Portsmouth, Sharyn Burns.

**Visualization:** Sharyn Burns.

**Writing – original draft:** Chimwemwe Tembo.

**Writing – review & editing:** Chimwemwe Tembo, Sharyn Burns.

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
