## [Decision Letter · Decision Letter 0]

20 Mar 2023

PGPH-D-23-00159

Postnatal depression and its social -cultural influences among adolescent mothers: A cross sectional study

Dear Dr. Tembo,

Thank you for submitting your manuscript to PLOS Global Public Health. After careful consideration, we feel that it has merit but does not fully meet PLOS Global Public Health’s publication criteria as it currently stands. Therefore, we invite you to submit a revised version of the manuscript that addresses the points raised during the review process.

We look forward to receiving your revised manuscript.

Kind regards,

Ahmed Waqas

Academic Editor

Journal Requirements:

Additional Editor Comments (if provided):

Reviewers' comments:

Reviewer's Responses to Questions

**Comments to the Author**

1. Does this manuscript meet PLOS Global Public Health’s publication criteria? Is the manuscript technically sound, and do the data support the conclusions? The manuscript must describe methodologically and ethically rigorous research with conclusions that are appropriately drawn based on the data presented.

Reviewer #1: Partly

Reviewer #2: Yes

2. Has the statistical analysis been performed appropriately and rigorously?

Reviewer #1: Yes

Reviewer #2: Yes

3. Have the authors made all data underlying the findings in their manuscript fully available (please refer to the Data Availability Statement at the start of the manuscript PDF file)?

Reviewer #1: Yes

Reviewer #2: Yes

4. Is the manuscript presented in an intelligible fashion and written in standard English?

Reviewer #1: No

Reviewer #2: Yes

5. Review Comments to the Author

Reviewer #1: Reviewer Report

Title: Postnatal depression and its social -cultural influences among adolescent mothers: A

cross sectional study

Date: 10/03/2023

This paper requires a lot of editorial works to improve the logical flow of information and coherence in the description of methods, results of the study, discussion, and recommendation. The draft also requires extensive work including correction of grammatical errors.

Despite these, the paper provides important information, and I would like to see it published if the authors could address some significant issues that require revision. I have detailed my concerns, comments, and questions below:

Abstract:

1. The information provided under background is exact copy of things written under the first three lines of the background in the main document of the manuscript. It should not be like this.

2. The reason (S) why the author (S) intended to conduct this study was not clearly stated in the background section

3. A cross-sectional survey was administered from September 2021 to March 2022….(Date should be specified)

4. Very important components of the methods like sample size, sampling technique, data entry methods… were missed.

5. On the conclusion you stated that, the prevalence of postnatal depression among adolescent mothers in Malawi is high. What was your reference or cut off point to describe as high or low?

Methods:

1. The information regarding defining the adolescent age range is not needed under study design and setting. It should better be included under operational definition or measurement.

2. The source and study populations were not clearly specified.

3. Criteria for inclusion was written as age greater than 20 under the population, are these age group your subject of interest?

4. One focus group discussion (FGD) with 15 adolescent mothers was conducted to determine the face validity of the survey. How can we use FGD to validate quantitative survey tool?

5. How was the tool developed? Adapted or adopted? What was the alpha value for internal consistency?

6. Data analysis was not well written

7. Ethical statement: lacks the following information

• Authors reporting experiments on humans must confirm that all methods were carried out in accordance with relevant guidelines and regulations.

• Must include a sentence confirming that informed consent was obtained from all subjects and/or their legal guardian(s).

• Here as the study also involves illiterate population, please ensure to provide an additional statement confirming that informed consent was obtained from their respective LARs as well.

Results

1. The calculated sample size was 395, but you invited 401 participants to participate in your study. Why do so? And your response rate became 99.5%. as to me you can’t invite more than the calculated sample size and your response rate should be 100% if you included 395 postnatal mothers. Try to differentiate non response rate, excluded participants and incomplete data/information.

2. Why not you include the first paragraph of the result section under subtitle sociodemographic chxs

3. The sum of the frequencies for some variables, such as polygamous marriage and marital age, does not equal 395.

4. According to Table 1, none of the participants have smoked or used tobacco in the last 12 months. As the other category is zero, it would be preferable to provide this information in text rather than a table.

5. The author (s) attempted to present factors linked with postnatal depression individually using chi square and logistic regression tables. It is preferable to combine both tables because the third table does not provide information on the strength and direction of the association.

6. What is the reason for the confidence interval being wide for the variables like IPV?

7. What parameters were considered for performing multivariable logistic regression analysis? Bivariate vs Multivariate analysis? confidence interval? COR?...

Discussion

1. What is your cut off point to say higher than, lower than and similar (inline)?

2. The authors justified the lack of studies on postnatal depression in Malawi, although in the discussion, it is said that the findings of this study are greater than the findings of two studies among postnatal adult women in Malawi in similar rural settings. This is perplexing!

Limitation

1. What do you think about the reliability and validity of Edinburgh Postpartum Depression Scale (EPDS)?

Reviewer #2: Overall, this is a well-written paper. However, the methods section (analysis) could be revised to reflect the results. As it is, the results are confusing because the reader is not aware of the analysis conducted. The discussion section is good. The author did well in comparing their findings with other studies. However, the paper could be made stronger by also including more implications on either policy or healthcare practices that can be incorporated to screen/ reduce adolescent postnatal depression.

See attachment for more comments.

6. PLOS authors have the option to publish the peer review history of their article (what does this mean?). If published, this will include your full peer review and any attached files.

**Do you want your identity to be public for this peer review?** For information about this choice, including consent withdrawal, please see our Privacy Policy.

Reviewer #1: No

Reviewer #2: No

---

## [Decision Letter · Decision Letter 1]

9 May 2023

PGPH-D-23-00159R1

Postnatal depression and its social -cultural influences among adolescent mothers: A cross sectional study

Dear Dr. Tembo,

Thank you for submitting your manuscript to PLOS Global Public Health. After careful consideration, we feel that it has merit but does not fully meet PLOS Global Public Health’s publication criteria as it currently stands. Therefore, we invite you to submit a revised version of the manuscript that addresses the points raised during the review process.

We look forward to receiving your revised manuscript.

Kind regards,

Ahmed Waqas

Academic Editor

Journal Requirements:

2. We have noticed that you have uploaded Supporting Information files, but you have not included a list of legends. Please add a full list of legends for your Supporting Information files after the references list. 

Additional Editor Comments (if provided):

Reviewers' comments:

Reviewer's Responses to Questions

**Comments to the Author**

1. If the authors have adequately addressed your comments raised in a previous round of review and you feel that this manuscript is now acceptable for publication, you may indicate that here to bypass the “Comments to the Author” section, enter your conflict of interest statement in the “Confidential to Editor” section, and submit your "Accept" recommendation.

Reviewer #1: All comments have been addressed

Reviewer #2: (No Response)

2. Does this manuscript meet PLOS Global Public Health’s publication criteria? Is the manuscript technically sound, and do the data support the conclusions? The manuscript must describe methodologically and ethically rigorous research with conclusions that are appropriately drawn based on the data presented.

Reviewer #1: No

Reviewer #2: Partly

3. Has the statistical analysis been performed appropriately and rigorously?

Reviewer #1: Yes

Reviewer #2: No

4. Have the authors made all data underlying the findings in their manuscript fully available (please refer to the Data Availability Statement at the start of the manuscript PDF file)?

Reviewer #1: Yes

Reviewer #2: Yes

5. Is the manuscript presented in an intelligible fashion and written in standard English?

Reviewer #1: Yes

Reviewer #2: Yes

6. Review Comments to the Author

Reviewer #1: all the comments have been addressed.

Reviewer #2: Good job making revisions. This information is much needed.

However, you still need to do some work on the methods and analysis section.

The abstract reports aOR which are adjusted odds ratios yet the study does not report any multivariable analysis. If only bivariate analyses were done, state the the rationale for not conducting multivariable analyses and how you would account for confounding variables.

7. PLOS authors have the option to publish the peer review history of their article (what does this mean?). If published, this will include your full peer review and any attached files.

**Do you want your identity to be public for this peer review?** For information about this choice, including consent withdrawal, please see our Privacy Policy.

Reviewer #1: **Yes: **Gemechu Gelan Bekele

Reviewer #2: No

---

## [Editor Report · Decision Letter 2]

17 May 2023

Postnatal depression and its social -cultural influences among adolescent mothers: A cross-sectional study

PGPH-D-23-00159R2

Dear Mrs Tembo,

We are pleased to inform you that your manuscript 'Postnatal depression and its social -cultural influences among adolescent mothers: A cross-sectional study' has been provisionally accepted for publication in PLOS Global Public Health.

Best regards,

Ahmed Waqas

Academic Editor